# Activation of Humoral Immunity during the Pathogenesis of Experimental Chronic Lung Allograft Dysfunction

**DOI:** 10.3390/ijms23158111

**Published:** 2022-07-23

**Authors:** Martin Reichert, Srebrena Atanasova, Kathrin Petri, Marian Kampschulte, Baktybek Kojonazarov, Gabriele Fuchs-Moll, Gabriele A. Krombach, Winfried Padberg, Veronika Grau

**Affiliations:** 1Laboratory of Experimental Surgery, Department of General and Thoracic Surgery, Excellence Cluster Cardio-Pulmonary Institute (CPI), Member of the German Center for Lung Research (DZL), Justus-Liebig-University Giessen, Feulgen-Strasse 10-12, D-35385 Giessen, Germany; srebrena.atanasova-koch@klinikum-hef.de (S.A.); kathrin.petri@chiru.med.uni-giessen.de (K.P.); gabriele.fuchs-moll@chiru.med.uni-giessen.de (G.F.-M.); winfried.padberg@chiru.med.uni-giessen.de (W.P.); veronika.grau@chiru.med.uni-giessen.de (V.G.); 2Department of Diagnostic and Interventional Radiology, University Hospital Giessen, Member of the German Center for Lung Research (DZL), Justus-Liebig-University Giessen, Klinikstrasse 33, D-35392 Giessen, Germany; marian.kampschulte@radiol.med.uni-giessen.de (M.K.); gabriele.krombach@radiol.med.uni-giessen.de (G.A.K.); 3Institute for Lung Health, Excellence Cluster Cardio-Pulmonary Institute (CPI), Member of the German Center for Lung Research (DZL), Justus-Liebig-University Giessen, Aulweg 130, D-35392 Giessen, Germany; baktybek.kojonazarov@innere.med.uni-giessen.de

**Keywords:** lung transplantation, allograft, chronic lung allograft dysfunction, CLAD, bronchiolitis obliterans, BOS, humoral immunity, B cells, alloreactive antibodies, autoreactive antibodies

## Abstract

Alloreactive and autoreactive antibodies have been associated with the development of chronic lung allograft dysfunction (CLAD), but their pathogenic role is disputed. Orthotopic left lung transplantation was performed in the Fischer-344 to Lewis rat strain combination followed by the application of ciclosporine for 10 days. Four weeks after transplantation, lipopolysaccharide (LPS) was instilled into the trachea. Lungs were harvested before (postoperative day 28) and after LPS application (postoperative days 29, 33, 40, and 90) for histopathological, immunohistochemical, and Western blot analyses. Recipient serum was collected to investigate circulating antibodies. Lung allografts were more strongly infiltrated by B cells and deposits of immunoglobulin G and M were more prominent in allografts compared to right native lungs or isografts and increased in response to LPS instillation. LPS induced the secretion of autoreactive antibodies into the circulation of allograft and isograft recipients, while alloreactive antibodies were only rarely detected. Infiltration of B cells and accumulation of immunoglobulin, which is observed in allografts treated with LPS but not isografts or native lungs, might contribute to the pathogenesis of experimental CLAD. However, the LPS-induced appearance of circulating autoreactive antibodies does not seem to be related to CLAD, because it is observed in both, isograft and allograft recipients.

## 1. Introduction

The median survival after lung transplantation is currently in the range of six years and is rather poor in comparison to other organs [1]. Chronic lung allograft dysfunction (CLAD) is the leading cause of allograft failure in the long term [1,2,3]. CLAD is an umbrella term describing different subforms of a chronic and persistent decline in pulmonary allograft function [1,4]. Essentially, three subforms have been described, obstructive bronchiolitis obliterans syndrome (BOS), which affects about two-thirds of all CLAD patients, restrictive allograft syndrome (RAS), and mixed forms combining features of BOS and RAS [1,4]. BOS is mainly characterized by fibrotic obliteration of bronchioles [1], while the fibrotic remodeling of the alveolar parenchyma is typical for RAS [1,5]. Mixed forms of BOS and RAS have also been described [1,4,6]. In addition, vascular changes, especially intimal hyperplasia and fibrosis of the adventitia, were found most commonly in CLAD lungs resulting in mild pulmonary hypertension [7,8,9]. The pathogenesis of CLAD is at best partially understood. It is thought to be a misguided regenerative response to pulmonary damage caused by a mixture of diverse factors such as ischemia/reperfusion injury, alloimmune reactions, infections, aspiration of gastric acid, and pollutants, which are all known risk factors for the development of CLAD [1,2,3]. It may be speculated, that the inevitable ischemia/reperfusion injury and alloimmunity cause an increased vulnerability of pulmonary transplants towards secondary harmful environmental or infectious stimuli, which eventually trigger CLAD.

There is an ongoing discussion on the role of donor-specific antibodies and autoreactive antibodies in CLAD [4,10,11,12,13,14]. Donor-specific antibodies often precede histopathologically defined acute antibody-mediated rejection episodes and the presence of donor-specific antibodies in recipient serum and/or graft tissue positively correlates with CLAD. The results of several, mainly uncontrolled studies aiming at the reduction or elimination of donor-specific or autoreactive antibodies were mixed, and there is a clear need for prospective clinical trials [10,13,14]. Autoreactive antibodies are detected alone or in combination with alloreactive antibodies and are often directed to k-α1 tubulin, collagen type V and vimentin [13,15,16,17]. The corresponding autoantigens seem to be demasked in allografts due to inflammatory processes caused by ischemia/reperfusion injury and acute rejection episodes [15,17]. Hence both, alloreactive and autoreactive antibodies, interact with antigens that are predominantly located in the allograft.

Experimental research on CLAD was hampered by a lack of meaningful experimental models [18]. We developed an experimental rat model that mirrors typical aspects of the pathogenesis and histopathology of human CLAD [19,20]. Left rat lungs are orthotopically transplanted in the Fischer-344 to Lewis rat strain combination that is characterized by a minor MHC class I mismatch. Early acute rejection is dampened by a short course of ciclosporine and the process of graft remodeling is triggered by a single intratracheal application of lipopolysaccharide (LPS) [19]. This experimental model combines two major risk factors for the development of CLAD, acute cellular rejection corresponding to stages A2/B1R to A3/B1R according to the nomenclature of the ISHLT (International Society for Heart and Lung Transplantation) [3,9], and additional rather unspecific pulmonary inflammation that is mediated by LPS [19,20]. LPS-induced inflammation does not only mimic bacterial infection, but similar mechanisms are expected to be activated by other risk factors for CLAD such as viral or fungal infection, aspiration of gastric fluid, or air pollutants [2,3,4]. In response to LPS instillation, rat allografts are infiltrated by leukocytes, and a protein-rich intraalveolar edema as well as severe endothelialitis develops [19]. This resembles histopathological changes in human lung allografts, which were associated with donor-specific antibodies [21,22,23]. In contrast to other experimental models involving orthotopic lung transplantation, which are hampered by a variable incidence of CLAD [18], CLAD develops in virtually all pulmonary allografts [19]. This experimental model offers the unique opportunity to differentiate changes in pulmonary allografts caused by inevitable factors, surgery, and alloimmunity, from “environmental” factors, in this case, LPS, which eventually trigger CLAD.

In this study, we test the hypothesis that instillation of LPS boosts the production of antibodies in experimental lung allograft recipients. We show, that application of LPS indeed results in increased B cell infiltration and in an increased deposition of antibodies in allografts but not in isografts. However, after treatment with LPS, autoreactive antibodies circulated in both, allograft and isograft recipients, which might question their contribution to the pathogenesis of CLAD.

## 2. Results

### 2.1. Graft Histopathology

Graft histopathology as well as graft infiltration by T cells and macrophages were described in detail before for this rat model for human CLAD [19,20]. In short, the histopathological changes on postoperative day 28 in pulmonary allografts are compatible with mild to moderate acute rejection. Instillation of LPS induced a strong leukocytic infiltration and a protein-rich intraalveolar edema within one day (day 29). These hallmarks of acute inflammation vanish during the following days (days 33 and 40). Initial signs of intimal hyperplasia as well as perivascular and peribronchiolar remodeling are seen as early as day 33 and are more pronounced on day 40 [19]. Full-blown CLAD, which seems to be a mixture of RAS and BOS, develops until day 90, which is characterized by severe bronchial and vascular remodeling as well as interstitial fibrosis [19]. Examples of the histopathology of pulmonary allografts and corresponding right native lungs of allograft recipients on days 40 and 90 are depicted in Figure 1.

### 2.2. B Cell Infiltration into Lungs

B cell infiltration was investigated by immunohistochemistry using monoclonal antibody OX33, directed to a pan-B cell marker. Pulmonary isografts and allografts before treatment with LPS (day 28) as well as one, five, and twelve days thereafter (days 29, 33, and 40) and five days after application of the control solution PBS (day 33) were investigated. Right native lungs were exclusively analyzed on day 33. Technical controls, in which the primary antibody was omitted, were virtually unstained. Immunopositive B cells were scarce in all isografts as well as in the right native lungs of both isograft and allograft recipients. In contrast, in all pulmonary allografts, B cell infiltrates were seen in perivascular and peribronchiolar regions, and after the application of LPS, B cells were also detected in the alveolar space (Figure 2a). For exploratory semi-quantification, a scoring system was used ranging from zero, corresponding to no increase in B cell density, to three, corresponding to the highest density of B cells observed. An allograft-specific increase in B cell density was seen in peribronchiolar (Figure 2b; *p* = 0.029, *n* = 4) and perivascular (Figure 2c; *p* = 0.029, *n* = 4) regions. Upon instillation of LPS (day 40 allografts) perivascular B cells tended to be more abundant compared to day 28 allografts (*p* = 0.086). The abundance of B cells in the alveolar space significantly increased in response to instillation of LPS but not to PBS. This increase was most prominent on postoperative day 40, twelve days after LPS instillation (Figure 2d; *p* = 0.029, *n* = 4). B cell abundance in right native lungs was very low (score 0) and remained unchanged after application of LPS (Appendix A).

### 2.3. Accumulation of Immunoglobulins in Lung Tissue

Accumulation of immunoglobulins (Ig) in lung tissue was first investigated by immunohistochemistry using antibodies directed to rat Ig and again, a scoring system ranging from zero to three was applied for exploratory semi-quantification. In isografts, allografts, and right native lungs from allograft recipients (Appendix A) immunoreactivity was prominent in almost all parts of the lung including the lumina of blood vessels (Figure 3a). Perivascular and peribronchiolar spaces as well as respiratory epithelia were only moderately stained in all isografts and allografts on postoperative day 28 (Figure 3a–e). At this time, immunopositive cells in the peribronchiolar/perivascular space, as well as inside alveoli, tended to be more abundant in allografts compared to isografts (*p* = 0.086 and *p* = 0.057, respectively). The situation changed upon instillation of LPS, which increased the abundance of Ig-positive intraalveolar cells in allografts compared to isografts within a single day (Figure 3a–e). Five and twelve days after LPS-instillation (postoperative days 33 and 40), peribronchiolar/perivascular and intraalveolar Ig-positive cells remained elevated in LPS-treated allografts compared to corresponding isografts (Figure 3a,c,e). Concomitantly, we observed an increased incidence of Ig deposits on the surface of respiratory epithelia (Figure 3a,d).

In a second approach, protein extracts of graft tissue (10 µg protein per lane) were separated on SDS polyacrylamide gels (10%) along with molecular mass marker proteins, and rat IgM or IgG were detected by Western blotting. The molecular masses of the resulting bands, ranging from about 60 to 75 kDa for IgM and 40 to 55 kDa for IgG, were in accordance with rat Ig heavy chains (Figure 4). A semi-quantitative exploratory densitometric analysis of the immunopositive bands revealed no increase in Ig immunoreactivity in isografts but a significant increase in both IgG and IgM reactivity in allografts on day 40 after transplantation (Figure 4; *p* = 0.029 versus day 40 isografts, *n* = 4 each).

### 2.4. C4d Deposits in Lung Tissue

C4d immunohistochemistry was performed on paraffin sections of normal Lewis lungs, isografts, and allografts on days 28, 29, 33, and 40 after transplantation. The intensity of the immunoreactivity was explored in a semi-quantitative manner using a scoring system (Figure 5). Again, in technical controls, in which the primary antibody was omitted, virtually no immunoreactivity was detected (Figure 5a). In the presence of the peptide that was used to raise the anti-C4d antiserum virtually no immunoreactivity was visible (Appendix A). A weak C4d immunoreactivity was seen on the luminal surface of the respiratory epithelium of isografts at all time points investigated (Figure 5). Instillation of LPS seemed to induce a moderate, transient increase in epithelial C4d immunoreactivity (Figure 5b). Blood vessels were devoid of immunoreactivity in isografts (Figure 5c). At most time points investigated, pulmonary allografts displayed C4d immunoreactivity on the surface of respiratory epithelia, and the staining intensity in allografts was stronger at day 40 compared to day 28 (Figure 5b). In addition to the staining on airway surfaces, vascular staining was seen almost exclusively in allografts (Figure 5a,c). However, in contrast to the well-described C4d staining on the luminal surface of vascular endothelial cells, which is a hallmark of antibody-mediated rejection [23], we saw C4d immunoreactivity in a line between the media and the adventitia in a localization corresponding to the external elastic lamina (Figure 5a). The vascular staining intensity was higher in day 40 allografts compared to day 28 allografts (Figure 5c). In the right native lungs, weak C4d immunoreactivity was detected on the respiratory epithelium and almost no vascular immunoreactivity was seen (Appendix A).

### 2.5. Antigens Recognized by Recipient Sera

Protein extracts from lung tissue isolated from healthy Fischer-344 (corresponding to allograft donor) and Lewis rats (corresponding to recipient and isograft donor) were separated by SDS polyacrylamide gel electrophoresis, and Western blots were performed using sera obtained from graft recipients at days 28, 29, 33 and 40 after transplantation. Secondary antibodies were directed either to rat IgM or to rat IgG. In controls, in which primary sera were omitted or in which sera from healthy untreated Lewis rats were used, neither IgM- nor IgG-immunoreactive bands were detected. The results of the Western blots are summarized in Table 1 and Table 2, and representative examples of the blots are depicted in Figure 6.

#### 2.5.1. Circulating IgM in Isograft Recipients

Sera from isograft recipients before intratracheal application of LPS (day 28) and shortly thereafter (day 29), were devoid of IgM immunoreactivity against pulmonary antigens. On day 33, autoreactive IgM antibodies to a 25 kDa antigen were detected in sera of all isograft recipients investigated (*n* = 6). In one of them, an additional autoreactive double-band was visible at a molecular mass of about 55 kDa. No IgM immunoreactivity was detectable in sera of day 40 isograft recipients (*n* = 4).

#### 2.5.2. Circulating IgM in Allograft Recipients

No IgM immunoreactivity was present in allograft recipients on days 28 and 29 after transplantation. On postoperative day 33, irrespective of allograft recipient treatment with vehicle or LPS, and on postoperative day 40, mixed results were obtained. Some of the sera were devoid of IgM immunoreactivity, in others specificities for autoantigens with molecular masses of about 25 kDa or 55 kDa were detected. Only in one day 40 allograft recipient autoreactive and alloreactive IgM antibodies to antigens at multiple molecular masses were detected (Table 1).

**Table 1 ijms-23-08111-t001:** Immunoreactivity of IgM of sera from lung transplant recipients on Western blots of protein extracts of Lewis and Fischer-344 rat lungs.

Postoperative Day, Graft	Intratracheal Treatment	*n*	Immunoreactivity
day 28, isograft	none	4	none
day 29, isograft	LPS	6	none
day 33, isograft	LPS	5 1	~25 kDa (autoreactivity) * ~25 kDa, ~55 kDa double-band (autoreactivity)
day 40, isograft	LPS	4	none *
day 28, allograft	none	4	none
day 29, allograft	LPS	6	none
day 33, allograft	vehicle	3	none
		1	~25 kDa (autoreactivity) *
		1	~55 kDa double-band (autoreactivity)
day 33, allograft	LPS	2 4	none ~25 kDa (autoreactivity)
day 40, allograft	LPS	4 1	~25 kDa (autoreactivity) multiple bands (autoreactivity and alloreactivity) *

* Shown as an example in Figure 6. LPS, lipopolysaccharide.

#### 2.5.3. Circulating IgG in Isograft Recipients

On day 28 after isogeneic transplantation, no circulating IgG antibodies interacted with pulmonary antigens (Table 2). Already one day after intratracheal instillation of LPS (day 29), circulating autoreactive IgG reacted with a double-band at an apparent molecular mass of about 55 kDa in all isograft recipients (*n* = 4). An additional autoreactive band at a molecular mass of about 35 kDa was seen in one of the blots. On postoperative day 33, similar results were obtained (*n* = 5), and a very strong background staining was obtained on postoperative day 40 (*n* = 4).

#### 2.5.4. Circulating IgG in Allograft Recipients

On day 28 after allogeneic transplantation, no circulating autoreactive or alloreactive IgG was detected in two recipients, whereas two other recipients displayed autoreactivity towards a band at an apparent molecular mass of about 50 kDa. All Western blots performed with day 29 sera showed a very strong background staining. In the serum of vehicle-treated allograft recipients on day 33 after transplantation, a mixed picture was seen ranging from no immunoreactivity, over autoreactivity towards a double-band at about 55 kDa, and a very strong background staining. In the serum of three day 40 allograft recipients, autoreactivity to a double-band at about 55 kDa was seen, whereas in one case multiple auto- and alloreactive specificities were detected (Table 2).
ijms-23-08111-t002_Table 2Table 2Immunoreactivity of IgG of sera from lung transplant recipients on Western blots of protein extracts of Lewis and Fischer-344 rat lungs.Postoperative Day,GraftIntratrachealTreatment*n*Immunoreactivityday 28, isograftnone5noneday 29, isograftLPS31~55 kDa double-band (autoreactivity)~55 kDa double-band, ~35 kDa (autoreactivity)day 33, isograftLPS5~55 kDa double-band (autoreactivity) *day 40, isograftLPS4strong background staining *day 28, allograftnone2none1~50 kDa (autoreactivity)1~55 kDa (autoreactivity)day 29, allograftLPS5strong background stainingday 33, allograftvehicle2none2~55 kDa double-band (autoreactivity)1strong background stainingday 33, allograftLPS1~55 kDa (autoreactivity)3~55 kDa double-band (autoreactivity)2>100 kDa, <10 kDa (autoreactivity) *1~55 kDa double-band, >100 kDa, <10 kDa (autoreactivity)day 40, allograftLPS31~55 kDa double-band (autoreactivity)multiple bands (autoreactivity and alloreactivity) ** Shown as an example in Figure 6. LPS, lipopolysaccharide.


## 3. Discussion

The presence of donor-specific and autoreactive antibodies correlates with human and experimental CLAD. It is, however, unclear, if these antibodies play a functional role or should be regarded as an epiphenomenon of other processes. As the pathogenesis of CLAD seems to be caused by a combination of alloreactivity and a second hit of rather unspecific pro-inflammatory stimuli, it is of interest to investigate changes in humoral immunity that are induced by the second hit. The aim of our study was to test the hypothesis that intratracheal instillation of LPS boosts humoral immunity specifically in allograft recipient rats. This hypothesis was largely confirmed, as more intraalveolar B cells and more Ig deposits were detected in allografts compared to isografts. However, circulating autoreactive Ig were induced by LPS in both, allograft and isograft recipients.

In comparison to the infiltration of pulmonary allografts by macrophages and T cells in the same experimental model [19], B cell infiltrates were less pronounced, which does not necessarily imply that B cells play a minor role in the pathogenesis of CLAD. B cell infiltrates in peribronchiolar and perivascular regions, seemed to depend on the presence of alloantigens because they were rare in isografts but present in all allografts, irrespective of intratracheal LPS or vehicle instillation. In contrast, the intraalveolar B cell number increased in allografts only after instillation of LPS. Hence, their infiltration seemed to be caused by the combination of alloimmunity and unspecific stimulation by LPS.

In the same line, immunohistochemistry using antibodies against all Ig classes revealed more immunoreactive cells in the alveolar space of allografts compared to isografts at all time points after instillation of LPS. Ig-immunopositive cells are most likely a mixture of B cells, cells displaying Fc receptors, and those displaying antigens that are detected by Ig of the host. On day 33, perivascular/peribronchiolar Ig-positive cells and Ig-positive respiratory epithelial cells were also more abundant in allografts compared to isografts. These observations were confirmed in Western blot experiments, which showed that Ig accumulated specifically in allografts on postoperative day 40 but not in isografts or right native lungs of allograft recipients. Hence, Ig accumulation seems to depend either on the presence of alloantigens or on preceding graft damage by acute rejection. More experiments are needed to identify the specificity of the antibodies accumulating in allografts.

Endothelial C4d immunoreactivity on 50% of interstitial capillaries is one of the criteria of antibody-mediated rejection in human allograft recipients [23]. No such endothelial staining was visible in the experimental isografts and allografts investigated here. Instead, C4d immunoreactivity was detected exclusively in allografts on the surface of bronchiolar epithelia and on elastic membranes of arteries. The staining of elastic fibers with antibodies to C4d is generally deemed to be unspecific. We conclude from these findings, that CLAD develops in this experimental model in the absence of endothelial C4d immunoreactivity on interstitial capillaries, a frequent aspect of antibody-mediated human allograft rejection.

The serum reactivity of isograft and allograft recipients was investigated on Western blots of protein extracts from healthy Lewis and Fischer-344 lungs. Intratracheal instillation of LPS seemed to induce circulating autoreactive IgM and IgG in both, isograft and allograft recipients, while alloreactive antibodies were rare. The only difference between isograft and allograft recipients might be, that in isograft recipients autoreactive IgM was transiently detected on postoperative day 33, while autoreactive IgM was also present in the serum of allograft recipients on postoperative day 40. We conclude from these findings that experimental CLAD can develop in the absence of circulating alloreactive antibodies. Sullivan et al. [24] described innate Th17 cells specific for collagen type V, k-α1-tubulin, and vimentin, autoantigens that are associated with the development of human [16] and experimental [25] CLAD. These cells are present in rodents and humans already before birth and kept in check by CD39^+^ regulatory T cells [24]. These Th17 cells and corresponding B cells might have been reactivated in response to LPS instillation. In the same line, a LPS-induced reduction of splenic mRNA expression of FoxP3, which is a typical transcription factor of regulatory T cells, was shown before in the same experimental model [20]. This might also explain our seemingly paradoxical observation that autoreactive IgG already circulated one day after instillation of LPS, whereas circulating autoreactive IgM was detected later. However, the presence of autoreactive antibodies in allograft and isograft recipients questions but does not formally exclude their role in the pathogenesis of CLAD.

Our study has numerous limitations. Large vascularized organ transplants can absorb autoreactive and alloreactive Igs and largely remove them from the circulation [12,26]. As already discussed, Igs of undefined specificity do indeed accumulate in allografts in response to LPS application. Hence, we cannot fully exclude the formation of alloreactive antibodies that were already absorbed by experimental allografts and not detected in recipient sera in Western blots. Another limitation is, that we did not identify the antigens detected by autoantibodies. Further, this study is not suited to answer the question, if the antibodies induced in response to instillation of LPS contribute to the pathogenesis of CLAD. To answer this question, more experimental studies are needed, in which B cells are specifically eliminated or fully inhibited in their function.

Another limitation of the study is, that the quantification of Ig accumulation in lung transplants should be regarded as semiquantitative. A quantification of immunohistochemical specimens is known to be just an estimation. Therefore, we performed Western blot experiments that should enable a more reliable densitometric quantification. It is common practice to include a housekeeping protein for data normalization. However, in the experimental model under investigation, this is impossible, because inflammation due to rejection or instillation of LPS dramatically changes the amount of protein-rich extracellular fluid and, hence, the relative concentration of intracellular housekeeping proteins. In addition, extracellular fluid such as intraalveolar edema contains plasma proteins including Ig, which might mask specifically accumulating Ig. Therefore, we adjusted the amount of proteins loaded onto the gels to 10 µg total protein and refrained from data normalization. The results seemed to prove us right as significant differences between isografts and allografts were seen only on day 40 after transplantation. This most probably reflects Ig accumulation and is not due to extracellular edema, which was shown to be most prominent on day 29, the day after LPS instillation, and gradually decreases thereafter [19].

Despite these limitations, we can conclude, that humoral immunity is boosted by LPS in pulmonary allograft recipient rats, resulting in increased alveolar B cell infiltration and increased deposition of Ig in the allograft tissue. These changes might contribute to the pathogenesis of CLAD because they are not observed to the same extent in isografts or in right native lungs from allograft recipients. However, LPS induced autoreactive circulating Ig in both, allograft and isograft recipients. The relevance of these antibodies for the pathogenesis of CLAD, however, deserves further research.

## 4. Materials and Methods

### 4.1. Experimental Animals and Lung Transplantation

Animal experiments were approved by the Regierungspräsidium Giessen, Germany (approval No. 49/2007 and No. 121/2014), they were performed in accordance with German animal protection laws and with the National Institutes of Health principles of laboratory animal care and we complied with the ARRIVE guidelines 2.0. Specified pathogen-free young adult male rats Lewis (LEW/OrlRj) rats were provided by Janvier Labs (ST Berthevin, France) and Fischer-344 (F344/DuCrl) by Charles River (Sulzfeld, Germany). Upon delivery, rats were kept in isolated ventilated cages and a day/night rhythm of 12 h, 21 to 22 °C, for at least one week before surgery and during the postoperative period. Rats had access to sterile standard chow and acidified water ad libitum. Both donor (F344 or LEW) and recipient (LEW) rats weighed 220 to 280 g at time of surgery.

The technique of experimental lung transplantation [27] and the experimental model for CLAD in the rat were described before [19,20]. In brief, orthotopic left lung transplantation was performed in the allogeneic Fischer-344 to Lewis rat strain combination, isogeneic transplantations were done in Lewis rats using a cuff technique for the anastomoses of blood vessels. Warm ischemic times remained below 18 min. Starting on the day of transplantation, graft recipients were treated with daily subcutaneous injections of ciclosporin (5 mg per kg body weight; Sandimmun^®^; Novartis Pharma, Nürnberg, Germany) for 10 days. During the fourth postoperative week, graft recipients were transiently anesthetized by inhalation of 4.5% isoflurane, and the technical success of transplantation was verified by standard computed tomography or micro-computed tomography (see below). Animals with non-aerated main bronchi and/or atelectatic lungs were excluded from the study. On postoperative day 28, graft recipients were transiently anesthetized by isoflurane and intratracheally instilled with 1 mL phosphate-buffered saline (PBS) per kg body weight as a control or with the same volume of PBS containing 0.5 mg lipopolysaccharide (LPS) per ml (L2654, *Escherichia coli* strain 026:B6, Sigma-Aldrich, Steinheim, Germany). Recipient rats were euthanized under deep isoflurane anesthesia (see below) at different time points after transplantation. Rats were exsanguinated via the caval vein to collect blood samples. Lungs were either fixed for histopathology or immunohistochemistry (see below) or cut in pieces and snap-frozen.

### 4.2. Perioperative Medication

Animals were anesthetized by inhalation of 4.5% isoflurane (Baxter, Unterschleissheim, Germany) followed by intraperitoneal injection of ketamine (90 mg per kg body weight; Ketavet, MSD Animal Health, Schwabenheim, Germany) and medetomidine (0.1 mg per kg bodyweight, Domitor, Orion Pharma, Espoo, Finland) as well as subcutaneous injection of glycopyrronium (15 µg per kg body weight; Robinul^®^, Riemser Pharma, Greifswald, Germany) and buprenorphine (0.2 mg per kg body weight; Buprenodale, Dechra, Northwich, UK). Only donor rats obtained an intravenous injection of heparin (1000 IU per kg body weight; Heparin-Natrium-ratiopharm^®^, Ulm, Germany). At the end of surgery, anesthesia was antagonized by subcutaneous injection of atipamezol (Antisedan, Orion Pharma) and another dose of buprenorphin (0.2 mg per kg body weight) was applied. A prophylactic single dose of 30 mg ampicillin (Ampicillin-ratiopharm^®^, Ratiopharm) was injected intraperitoneally.

### 4.3. Standard Computed Tomography

Cross-sectional imaging was performed on a MSCT Somatom Force (Siemens, Forchheim, Germany). The technical parameters of the examination protocol were as follows: collimation-width, 0.6 mm; pitch factor, 0.75; volume-weighted computed tomography dose index (CTDIvol), 12.5 mGy; tube voltage: 110 kVp. Image reconstruction was performed with an edge-enhancing convolution kernel (B157s). The reconstructed field of view (FOV) was 8 cm × 8 cm with an image matrix of 512 × 512 pixels and an overlapping slice thickness of 0.6 mm. Image acquisition was performed in the lung window (WL -40 HU/ WW 1500 HU) and subsequently under individual windowing (PACS, INFINITT PACS, INFINITT Europe, Frankfurt, Germany).

### 4.4. Micro-Computed Tomography

Images were acquired using a Quantum GX microCT scanner (PerkinElmer, Inc., Waltham, MA, USA). Rats were placed on a scanner bed, which was translated longitudinally to align the animal chest within the center of the field of view. The scanner’s complementary metaloxide-semiconductor X-ray flat-panel detector was set to allow image acquisition with an X-ray tube voltage of 90 kV and a current of 80 μA. Micro-CT data were collected in list-mode over a single complete gantry rotation with a total rotation time of 4 min (14,688 frames collected in total). Raw projection images were processed using a proprietary algorithm for intrinsic retrospective respiratory gating and then reconstructed using a filtered back-projection algorithm on a dedicated graphics processing unit.

### 4.5. Histopathology and Immunohistochemistry

Lungs were fixed in situ by intratracheal instillation of 4% freshly dissociated buffered paraformaldehyde followed by immersion for 24 h in the same solution and embedded in paraffin. An Olympus BX51 microscope (Olympus, Hamburg, Germany), combined with the Axiocam 305 color camera (Carl Zeiss, Jena, Germany) and the Zen 2.3 blue edition software (Carl Zeiss) were used for evaluation.

For histopathological evaluation, sections of 6 to 8 µm thickness were dewaxed, rehydrated, and stained with hemalum eosin (H&E), acidic orcein, or Heidenhain´s AZAN.

For immunohistochemical detection of B cells and C4d dewaxed and rehydrated sections (6 to 8 µm) were stained without pretreatment. Before detection of Ig, slides were treated with 0.5 mg/mL Protease Type XIV (Sigma-Aldrich), 50 mM Tris-HCl buffer, pH 7.6, 0.9% NaCl, for 15 min at room temperature. Endogenous peroxidase activity was blocked with 1% H_2_O_2_ in PBS for 30 min. After washing in PBS, pH 7.2, the sections were incubated for 30 min either with PBS, pH 7.2, 1% BSA (Serva, Heidelberg, Germany) for detection of Ig or with PBS, pH 7.2, 1% BSA, and 0.1% NaN_3_ (p.a., Merck, Darmstadt, Germany) for the detection of B cells and C4d, followed by over-night incubation with an appropriate dilution of primary antibodies in the same solution at 4 °C. Bound monoclonal antibody OX33, directed to the B cell-specific antigen CD45RA (1:3000; Bio-Rad, Hercules, CA, USA) and polyclonal rabbit anti-rat C4d (1:50; Hycult Biotech, Wayne, PA, USA) were detected by horseradish peroxidase-labeled anti-mouse or anti-rabbit EnVision^®^ (Dako by Agilent, Santa Clara, CA, USA) containing 5% heat-inactivated normal rat serum. Rat Ig was detected by horseradish peroxidase-conjugated polyclonal rabbit anti-rat Ig (1:50; Dako by Agilent). On control sections, which were included in each experiment, the primary antibody was omitted. Additionally, to control for the specificity of the antibody, the peptide used for raising the antibody to C4d (STPAPRNPSEPVPQ, thinkpepdides, Oxford, UK) was added to the primary antibody in the concentration of 20 µg/500 µL in PBS, 1% BSA, 0.1% NaN_3_. Peroxidase activity was visualized by 3,3′-diaminobenzidine (DAB, Sigma-Aldrich) and sections were lightly counterstained with hemalum. B cell infiltrates were scored separately in peribronchiolar, perivascular and alveolar areas. The intensity of extracellular Ig staining was scored in peribronchiolar/perivascular regions and on the surface of the respiratory epithelium. The abundance of Ig-positive cells was evaluated in peribronchiolar/perivascular regions and in alveolar spaces. C4d immunoreactivity was scored on the respiratory epithelium and in arterial walls.

### 4.6. Electrophoresis and Western Blotting

The protein content of lung extracts was measured using a bicinchoninic acid kit (Micro BCA protein assay, Thermo Fisher Scientific, Waltham, MA, USA). Extracts (10 µg protein per lane) were separated by SDS polyacrylamide gel electrophoresis (SDS-PAGE, 10%) along with dual color precision plus protein standards (Bio-Rad, Hercules, CA, USA) and transferred to Immobilon^®^-P polyvinylidene difluoride membranes (Merck Millipore, Burlington, MA, USA). PBS, 0.01% Tween-20 was used for most washing steps and PBS without detergent before the visualization of bound horseradish peroxidase using SuperSignal West Dura Extended Duration Substrat (Thermo Fisher Scientific) for detection of Ig content of transplants or Lumi-Light substrate (Roche, Mannheim, Germany) for detection of reactivity of recipient sera and High Performance Chemiluminescence Films (GE Healthcare, Chicago, IL, USA).

When the Ig content of transplants and native right recipient lungs was investigated, unsaturated protein binding sites on the membranes were blocked with 2.5% skimmed milk powder (Carl Roth, Karlsruhe, Germany) in PBS. Blots were probed with horseradish peroxidase-conjugated goat antibodies to rat IgM (1:25,000; No. 31476, Thermo Fisher Scientific) or IgG (1:5000; No. 31475, Thermo Fisher Scientific). One sample each of isografts, allografts, and right native lungs from allograft recipients of postoperative days 28, 29, 33, and 40 were grouped together on one blot. Blots were analyzed densitometrically using the Alpha Digidoc 1201 software (Alpha Innotech Co., San Leandro, CA, USA). The intensity of the individual background was subtracted from the intensity of selected immunoreactive bands. The results of densitometry were normalized to the signals obtained for allografts on postoperative day 40.

To analyze the reactivity of recipient sera, membranes were blocked with 2.5% skimmed milk powder in PBS for 60 min and probed with 1:200 dilutions of recipient serum in PBS plus Roti^®^block (Carl Roth) over-night at 4 °C. Bound rat antibodies were detected with horseradish peroxidase-conjugated goat antibodies to rat IgM (1:50,000; No. 31476, Thermo Fisher Scientific) or IgG (1:5000; No. 31475, Thermo Fisher Scientific). In each experiment, the following negative controls were included: omission of the primary serum and use of serum from a healthy untreated Lewis rat. Mouse anti-β-actin antibodies (clone A2228, Sigma-Aldrich) were used to control for equal loading. To ensure better comparability, the same serum from an allograft recipient (day 33, LPS-treated recipient) was repeatedly included in all sets of blots that were processed together as the positive control.

### 4.7. Statistical Analysis

Statistical analyses were performed using GraphPad Prism (Version 9 for Windows, GraphPad Software, San Diego, CA, USA, www.graphpad.com). Multiple groups were initially analyzed with Kruskall–Wallis test for global effects. If applicable, i.e., in cases of *p* ≤ 0.05 in Kruskall–Wallis test, the unpaired, two-tailed Man–Whitney U test was performed for comparisons between individual groups; *p* ≤ 0.05 was considered to indicate statistical significance.

## Figures and Tables

**Figure 1 ijms-23-08111-f001:**
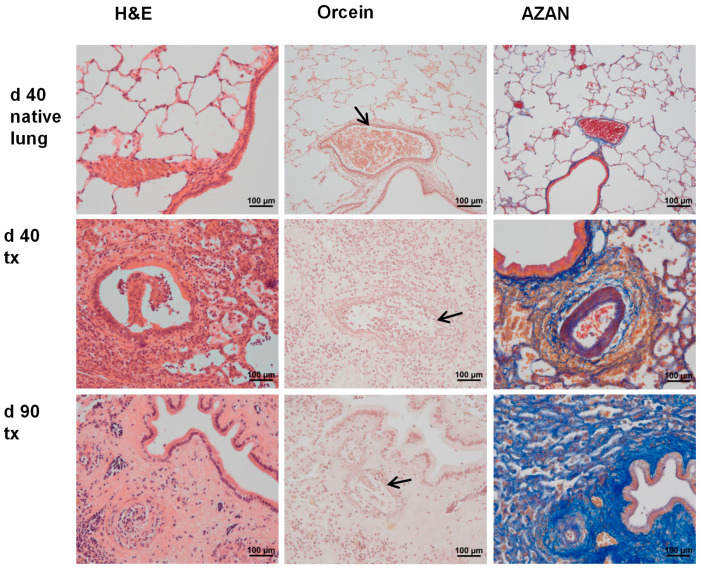
Histopathology of native lungs and allografts (tx). Representative paraffin sections stained with hemalum and eosin (H&E), acidic orcein, or Heidenhain’s AZAN are shown (*n* ≥ 4 each). Experimental animals were euthanized 40 and 90 days after transplantation (d 40, d 90). Arrows are pointing to the internal elastic membrane of arteries, which are visualized by orcein. Extracellular matrix is stained in blue after AZAN staining.

**Figure 2 ijms-23-08111-f002:**
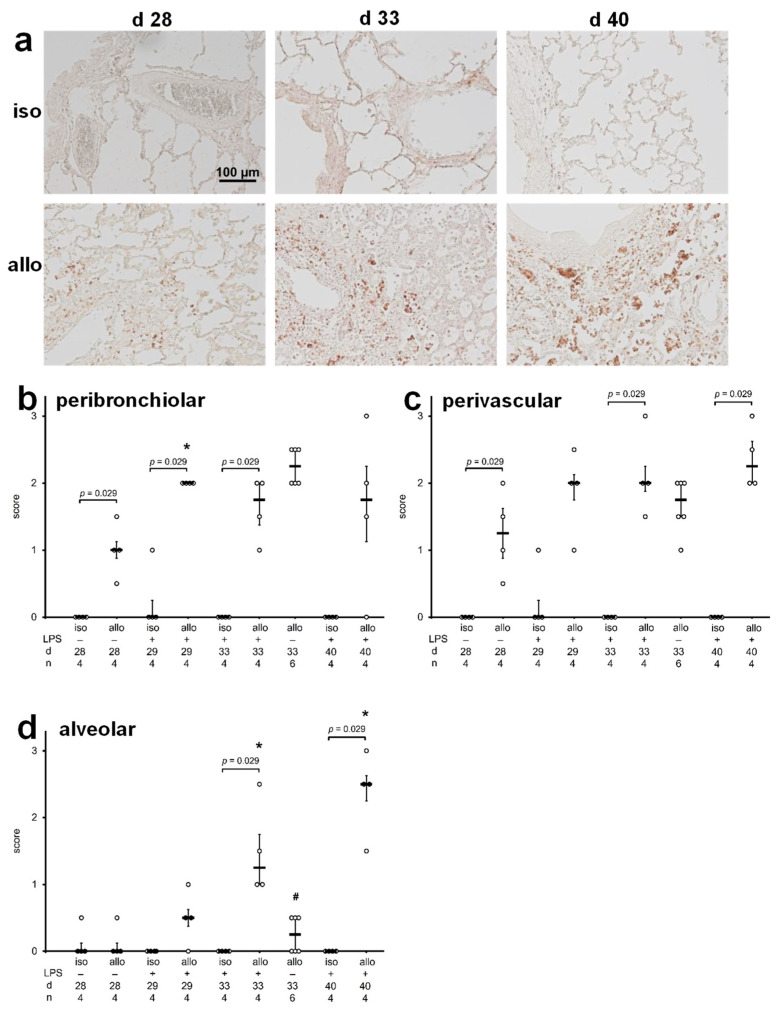
Graft infiltration by B cells. Monoclonal antibody OX33 was used to detect B cells by immunohistochemistry on paraffin sections of pulmonary isografts (iso) and allografts (allo). Grafts were investigated on postoperative day 28 (d 28) before intratracheal instillation of lipopolysaccharide (LPS) or phosphate-buffered saline (PBS), as well as on postoperative days 29 (d 29), 33 (d 33), and 40 (d40), 1, 5 and 12 days after treatment with LPS, respectively. Immunopositive cells are stained in brown, and cell nuclei are lightly counterstained with hemalum (**a**). The intensity of B cell infiltration in peribronchiolar (**b**) perivascular (**c**) and alveolar (**d**) regions was evaluated using a scoring system ranging from 0 for virtually no B cell infiltrates to 3, the strongest infiltrates observed. Data are presented as individual data points, bar represents median, whiskers encompass the 25th to 75th percentile. Explorative data analysis was performed using Kruskal–Wallis test followed by Mann–Whitney U-test; * *p* ≤ 0.05, different from allo d 28, # *p* ≤ 0.05 allo d 33 PBS versus allo d 33 LPS.

**Figure 3 ijms-23-08111-f003:**
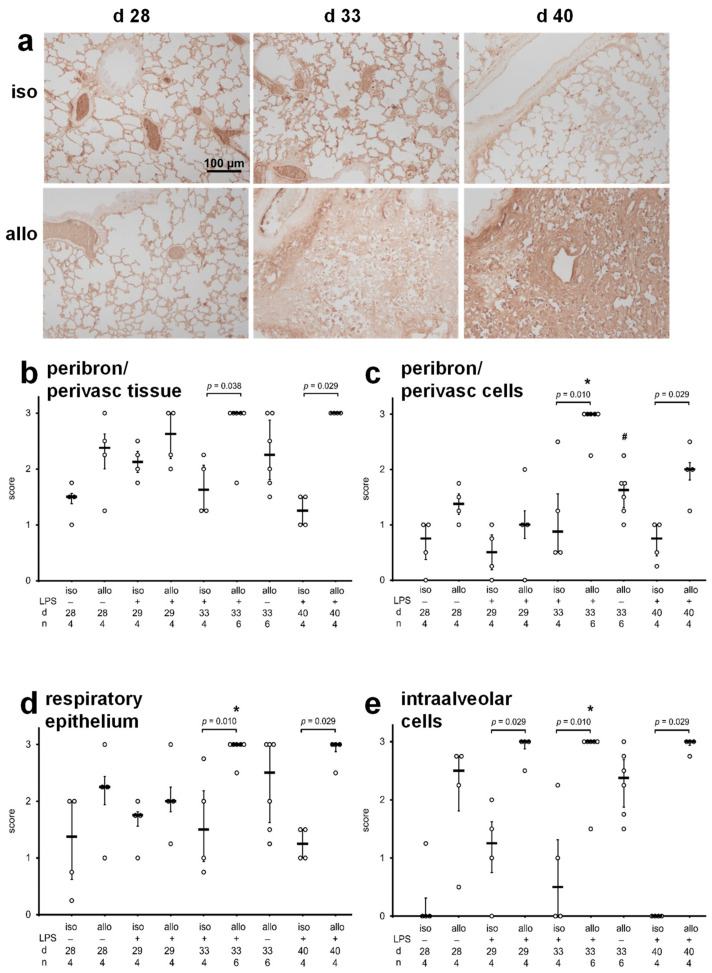
Accumulation of immunoglobulins (Ig) in lung tissue. Antisera directed to rat Ig were applied to paraffin sections of pulmonary isografts (iso) and allografts (allo). Grafts were investigated on postoperative day 28 (d 28) before intratracheal instillation of lipopolysaccharide (LPS) or phosphate-buffered saline (PBS), as well as on postoperative days 29 (d 29), 33 (d 33), and 40 (d 40), 1, 5 and 12 days after treatment with LPS, respectively. Immunopositive regions and cells are stained in brown, and cell nuclei are lightly counterstained with hemalum (**a**). The intensity of Ig deposits in peribronchiolar/perivascular (peribron/perivasc) tissue (**b**), the abundance of Ig-positive cells in peribronchiolar/perivascular regions (**c**), the intensity of Ig deposits on the luminal surface of the respiratory epithelium (**d**), and the abundance of Ig-positive cells in alveoli (**e**) were evaluated using a scoring system ranging from 0 for virtually no immunoreactivity to 3, the strongest immunoreactivity observed. Data are presented as individual data points, bar represents median, whiskers encompass the 25th to 75th percentile. Explorative data analysis was performed using Kruskal–Wallis test followed by Mann–Whitney U-test; * *p* ≤ 0.05, different from allo d 28, # *p* ≤ 0.05 allo d 33 PBS versus allo d 33 LPS.

**Figure 4 ijms-23-08111-f004:**
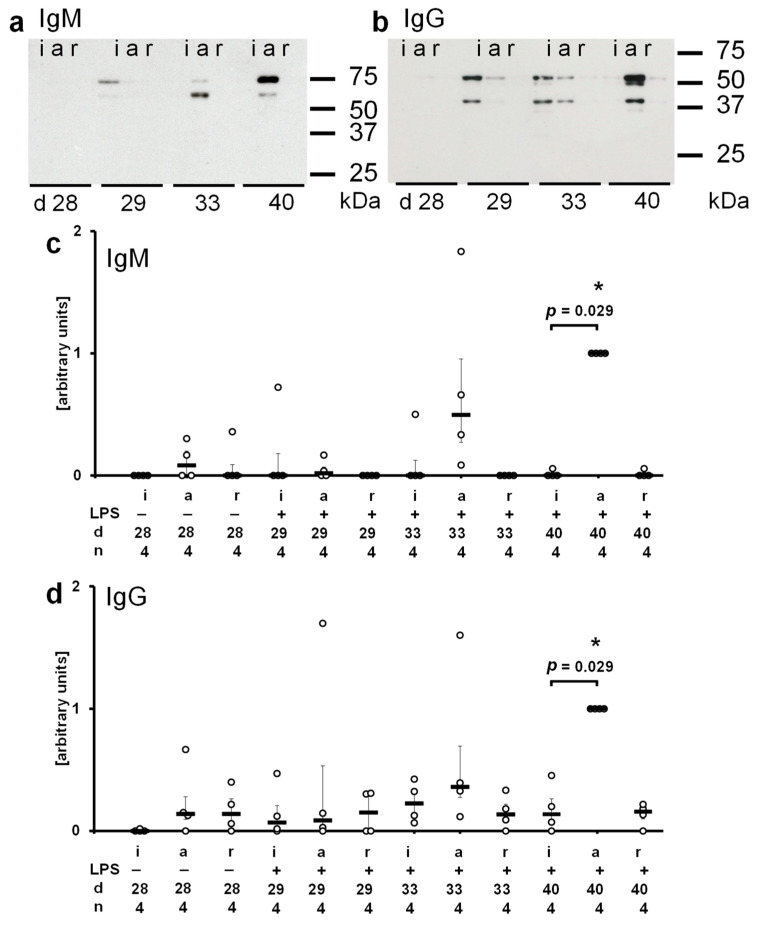
Accumulation of immunoglobulin (Ig) M and IgG in lung tissue. Protein extracts of pulmonary isografts, allografts, and right native lungs of allograft recipients (i, a, r) were separated on SDS polyacrylamide gels (10%) along with molecular mass standards. Proteins were blotted and probed with antibodies directed to rat IgM or rat IgG. Postoperative day 28 (d 28) before intratracheal instillation of lipopolysaccharide (LPS) as well as on postoperative days 29 (d 29), 33 (d 33), and 40 (d 40), 1, 5, and 12 days after treatment with LPS, respectively, were investigated. Representative Western blots for IgM (**a**) and IgG (**b**) are depicted (*n* = 4 each). A semiquantitative, densitometric analysis was performed and the results are depicted for IgM and IgG in (**c**) and (**d**), respectively. Data are presented as individual data points, bar represents median, whiskers encompass the 25th to 75th percentile. Explorative data analysis was performed using Kruskal–Wallis test followed by Mann–Whitney U-test; * *p* ≤ 0.05, different from d 28 allografts.

**Figure 5 ijms-23-08111-f005:**
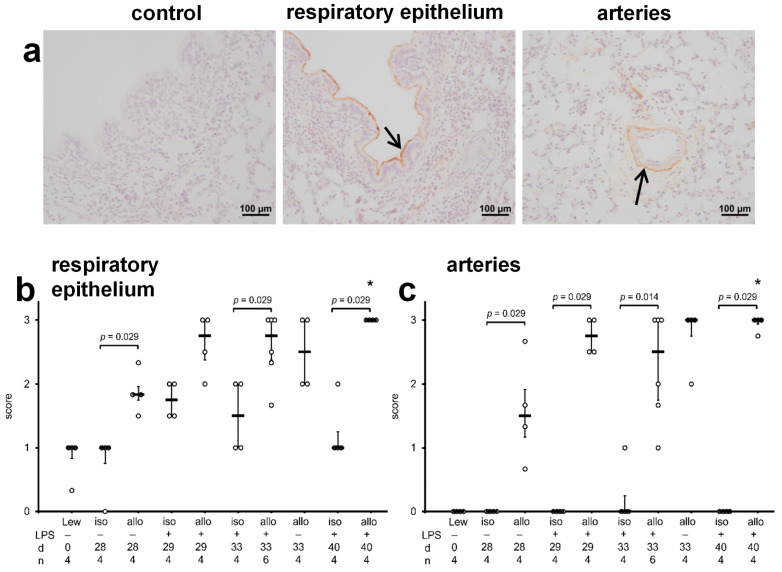
Deposition of C4d in lung tissue. C4d was detected by immunohistochemistry on paraffin sections of pulmonary isografts (iso) and allografts (allo). Grafts were investigated on postoperative day 28 (d 28) before intratracheal instillation of lipopolysaccharide (LPS) as well as on postoperative days 29 (d 29), 33 (d 33), and 40 (d 40), 1, 5, and 12 days after treatment with LPS, respectively. Immunopositive regions are stained in brown, and cell nuclei are lightly counterstained with hemalum. Examples of sections of d 40 allografts are depicted (**a**). In the section labeled control, the primary antibody was omitted (**a**). Further, examples showing C4d deposits on the respiratory epithelium (arrow) and on the arterial wall at the interface between media and intima (arrow) are depicted (**a**). The intensity of the C4d immunoreactivity on the respiratory epithelium and in the arterial wall was evaluated using a scoring system ranging from 0 for virtually no immunoreactivity to 3, the strongest immunoreactivity observed (**b**,**c**). Data are presented as individual data points, bar represents median, whiskers encompass the 25th to 75th percentile. Explorative data analysis was performed using Kruskal–Wallis test followed by Mann–Whitney U-test; * *p* ≤ 0.05, different from allo d 28.

**Figure 6 ijms-23-08111-f006:**
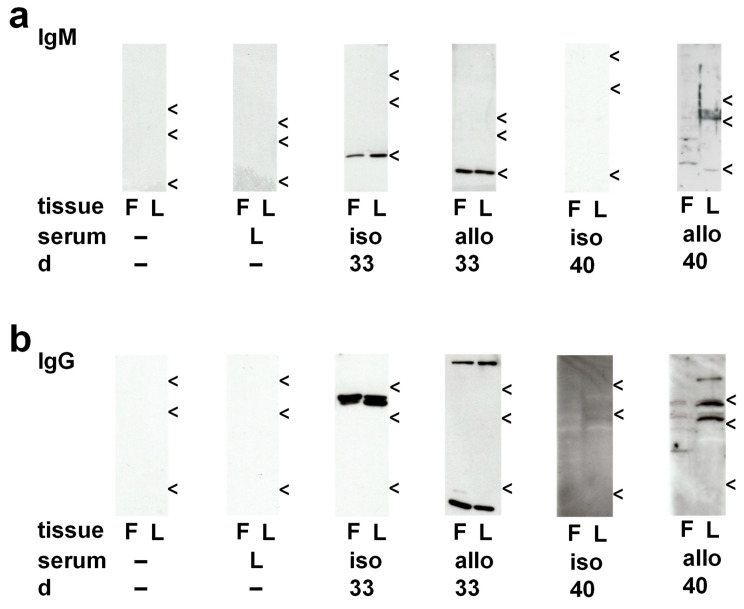
Immunoreactivity of sera from lung transplant recipients on Western blots of protein extracts of native Fischer-344 (F) and Lewis (L) rat lungs. Representative examples showing immunoreactivity of IgM (**a**) and IgG (**b**) of sera from native Lewis rats (L) and isogeneic (iso) or allogeneic (allo) lung transplant recipients on postoperative days 33 (d 33) and 40 (d 40), 5 and 12 days after treatment with LPS, respectively. Arrows indicate molecular masses: 75 kDa, 50 kDa, 25 kDa in top to bottom order.

## Data Availability

The data presented in this study are available on request from the corresponding author.

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
