# Peer review of "Activation of Humoral Immunity during the Pathogenesis of Experimental Chronic Lung Allograft Dysfunction"

_ijms, 2022, doi:10.3390/ijms23158111_

Round 1
Reviewer 1 Report
Martin Reichert and colleagues studied the role of humoral immunity in a well-defined orthotopic lung transplantation model in the rat. Upon LPS stimulation, which simulates an unspecific second hit and leads to CLAD, lung allografts were strongly infiltrated by B cells and deposits of immunoglobulin G and M were more prominent allografts compared to native lungs. Autoreactive antibodies could not be associated with experimental CLAD and circulating alloreactive antibodies were hardly detected, possible due to the absorption to the graft.
The study is well performed and is an important step for further studies on the role of humoral immunity on CLAD development. Limitations are thoroughly discussed by the authors.
I have only minor comments:
-Page 2, line 54/55: Please refer to original research describing the mixed phenotype. A mixed phenotype and its impact on outcome have been in detailed described: Leuschner et al. JHLT 2020; doi: 10.1016/j.healun.2020.08.008.
-Even though C4d staining has been proposed by the pathology council of the ISHLT, due to its week correlation with antibody mediated rejection in human lung transplantation its use is debated (Kauke T et al. 2015; DOI: 10.1111/tan.12626). Therefore, I am not surprised that CLAD develops in this experimental model in the absence of C4d staining. I would not call it a typical hallmark of AMR (Page 14, line 359)
Author Response
-Page 2, line 54/55: Please refer to original research describing the mixed phenotype. A mixed phenotype and its impact on outcome have been in detailed described: Leuschner et al. JHLT 2020; doi: 10.1016/j.healun.2020.08.008.
--> We have included the reference of original research on different types of CLAD including the mixed phenotype at this point.
-Even though C4d staining has been proposed by the pathology council of the ISHLT, due to its week correlation with antibody mediated rejection in human lung transplantation its use is debated (Kauke T et al. 2015; DOI: 10.1111/tan.12626). Therefore, I am not surprised that CLAD develops in this experimental model in the absence of C4d staining. I would not call it a typical hallmark of AMR (Page 14, line 359)
--> Thank you for this comment, we have modified this controversial statement in the manuscript.
Reviewer 2 Report
Nice work demonstrating the role of humoral immunity in the development of CLAD in the animal model of allotransplantation following LPS instillation.
Author Response
Nice work demonstrating the role of humoral immunity in the development of CLAD in the animal model of allotransplantation following LPS instillation.
--> Thank you for this comment.